

# Real time anatomical landmarks and abnormalities detection in gastrointestinal tract

Zeshan Khan and Muhammad Atif Tahir

FAST School of Computing, National University of Computer and Emerging Sciences, Islamabad, Karachi, Sindh, Pakistan

## ABSTRACT

Gastrointestinal (GI) endoscopy is an active research field due to the lethal cancer diseases in the GI tract. Cancer treatments result better if diagnosed early and it increases the survival chances. There is a high miss rate in the detection of the abnormalities in the GI tract during endoscopy or colonoscopy due to the lack of attentiveness, tiring procedures, or the lack of required training. The procedure of the detection can be automated to the reduction of the risks by identifying and flagging the suspicious frames. A suspicious frame may have some of the abnormality or the information about anatomical landmark in the frame. The frame then can be analysed for the anatomical landmarks and the abnormalities for the detection of disease. In this research, a real-time endoscopic abnormalities detection system is presented that detects the abnormalities and the landmarks. The proposed system is based on a combination of handcrafted and deep features. Deep features are extracted from lightweight MobileNet convolutional neural network (CNN) architecture. There are some of the classes with a small inter-class difference and a higher intra-class differences, for such classes the same detection threshold is unable to distinguish. The threshold of such classes is learned from the training data using genetic algorithm. The system is evaluated on various benchmark datasets and resulted in an accuracy of 0.99 with the F1-score of 0.91 and Matthews correlation coefficient (MCC) of 0.91 on Kvasir datasets and F1-score of 0.93 on the dataset of DowPK. The system detects abnormalities in real-time with the detection speed of 41 frames per second.

# INTRODUCTION

Gastrointestinal (GI) cancer significantly contributes to mortality among various cancer types. Colon cancer is fifth in the most dangerous cancer types concerning the number of affected patients and deaths due to cancer. Rectum cancer is another type of GI tract cancer that is the 8th most dangerous cancer type in terms of number of cancer patients in the last five years (*Sung et al., 2021*). The mortality rate of cancer patients is higher in Asia compared to other countries (*Sung et al., 2021*), and one of the reasons for death due to cancer is its late identification. Cancer becomes mortal when it becomes symptomatic; therefore, there is a need for screening of the high prevailing areas of infection to reduce

Corresponding author
Zeshan Khan,
zeshan.khan@nu.edu.pk

the risk of cancer by early detection even before it becomes symptomatic. The endoscopic procedures are the gold standard for the detection of GI abnormalities and cancers by the analysis of the whole GI tract, especially the stomach and the large bowel (CRC) due to its high vulnerability towards infection (*Riegler et al., 2017*; *Pogorelov et al., 2018*). The medical expert needs much time to diagnose the abnormalities early for cancer detection or detection of cancer-causing abnormalities. There is a need for a computer-aided diagnostic system to quickly detect abnormalities to minimize the intervention of the human medical expert (*Suzuki, 2012*; *Doi, 2007*).

There are many techniques to analyze GI tract images or videos using computer-aided diagnostic (CAD) systems. Most CAD systems primarily rely on machine learning and image processing techniques. The systems working on the machine learning algorithms on the texture and color features of the images are faster in detection. However, the detection score of such systems is lower than computer vision-based systems. The deep learning-based systems are sound in detection with a slow detection speed. There is a need for a system that offers a high detection speed in real-time and a justifiable detection accuracy.

This research presents a three-step model for detecting abnormalities in the GI tract images. The first step relies on data preprocessing to address two critical challenges in the GI tract datasets. The first challenge is of light reflection on the images due to endoscopic procedures which is mitigated by applying the Open-CV Telea method. The second challenge in the endoscopic datasets is the high-class imbalance, which is mitigated with the generation of new images for all the classes so that the argument and original images combined make the dataset balanced. The following research step is to extract the most suitable features to represent images better. For this purpose, various features used in the literature have been explored, and a set of best features is selected after applying various feature combinations. The misleading features are excluded using feature accuracies and diversities. The third step of the research is to compute classes for an instance, which has further been divided in two steps. In the first part, a neural network is used to compute the probability of each class for an image and then to select the class based on different thresholds for each class instead of a 0.5 threshold or max value. The threshold for each class is computed by applying a genetic algorithm with the random initial threshold for all the classes. The methodology is named LiRE-CNN, as it uses some of the Lire features in combination with a neural network architecture.

The detection algorithm of the LiRE-CNN is applied to different benchmark datasets with several modifications in the detection approach. The algorithm is also applied with and without preprocessing. The final results show that applying the preprocessing of reflection removal and augmentation and the neural network can achieve the best accuracy with optimal time. The texture features and local binary patterns are faster commutable, and the deep features are good deciders. Combining both types of features resulted in an accuracy of 0.99 with a detection speed of 41 frames per second. The same algorithm resulted in the F1-score of 0.91, 0.90, and 0.91 on the Kvasir versions V1 (*Pogorelov et al., 2017c*), V2 (*Pogorelov et al., 2018*), and V3 (*Borgli et al., 2020*), respectively, with the same 41 FPS.

This article is organized as follows. 'Related Work' discusses the related work. The proposed methodology is presented in 'Proposed Approach'. 'Experimental Setup' describes the experimental set-up followed by a discussion about results in 'Results and Discussion'. 'Conclusion and Future Work' concludes the article.

## RELATED WORK

The primary objective of the endoscopy or colonoscopy is to detect the functionality of the Gastro-Intestinal tract. Several possible abnormalities in the GI tract include polyps, lesions, esophagitis, and ulcerative colitis. There is a strong disagreement among medical specialists about the decision of the various abnormalities that can cause low detection accuracy of the issues in GI tract. Intense work is available on the detection, classification, localization, and segmentation of various diseases in endoscopy.

The detection tasks are mainly done for the detection of polyps and lesions using elliptical shape, color, position, texture, local intensity variation patterns, and global geometric constraints as features (*Hwang et al., 2007*; *Alexandre, Nobre & Casteleiro, 2008*; *Tajbakhsh et al., 2014*).

The primary issue in medical diagnostics is the limited availability of labeled datasets for applying machine learning algorithms. The limited dataset availability also generates a problem of class imbalance in the case of detection. It may lead to a higher number of non-polyps images than polyps.

The data augmentation is done by using three major techniques.

1. Generative adversarial network (GAN): The GANs are generative models used for the generation of images from some of the existing images of various classes.
2. Angular flips: The angular flips have been used for the data augmentation with the operations of the rotation and flipping image with various angles, cropping, and resizing of the image (*Luo et al., 2019*; *Chang et al., 2019*).
3. Noise: Adding noise is another approach used for the data augmentation by various authors (*Chang et al., 2019*; *Khan & Tahir, 2018*). The noise can be added in various randomized techniques, including contrast, brightness, gamma, blur, motion blur, and the invention of image and scale change (*Chang et al., 2019*).

The detection of the abnormalities is done by several approaches including deep learning and machine learning and a combination of both the approaches (*Naqvi et al., 2017*; *Tan & Triggs, 2010*; *Khan & Tahir, 2018*; *Krizhevsky, Sutskever & Hinton, 2012*; *He et al., 2016*; *Zeiler & Fergus, 2014*; *Simonyan & Zisserman, 2014*; *Szegedy et al., 2015*; *Huang et al., 2017*).

Some authors presented approaches of the deep neural network as a feature extractor and then the classification of the images on those deep features with or without state-of-the-art features (*Naqvi et al., 2017*; *Zhang et al., 2016*).

There is a sound contribution available on the detection of polyps, cancer, or lesions in the GI tract using various image features (*Ojala, Pietikäinen & Harwood, 1996*; *Liu et al., 2016*; *Liao, Law & Chung, 2009*; *Zhu, Bichot & Chen, 2010*).

The approaches using these set of features with the classification of support vector machine (SVM) (*Hearst et al., 1998*), random forest, and the k-nearest neighbors (KNN)

**Table 1  Analysis of several approaches for binary class classification on various polyps or lesion detection datasets.**

| Sr. No. | Evaluation measure | Best scores | Dataset |
| --- | --- | --- | --- |
| 1 | Sensitivity | 0.94 (*Deeba et al., 2018*) | CE Bleeding Dataset (*Deeba et al., 2018*; *Faigel & Cave, 2008*) |
| 2 | Specificity | 0.95 (*Deeba et al., 2018*) | CE Bleeding Dataset (*Deeba et al., 2018*; *Faigel & Cave, 2008*) |
| 3 | Precision | 0.98 (*Zhao et al., 2021*) | WCE-279 Bleeding Dataset (*Zhao et al., 2021*) |
| 4 | Recall | 0.98 (*Zhao et al., 2021*) | WCE-279 Bleeding Dataset (*Zhao et al., 2021*) |
| 5 | F1-score | 0.98 (*Zhao et al., 2021*) | WCE-279 Bleeding Dataset (*Zhao et al., 2021*) |
| 6 | Accuracy | 0.98 (*Zhao et al., 2021*) | WCE-279 Bleeding Dataset (*Zhao et al., 2021*) |
| 7 | ROC | 0.95 (*Alexandre, Nobre & Casteleiro, 2008*) | Polyp2007 (*Alexandre, Nobre & Casteleiro, 2008*) |
| 8 | AUC | 0.83 (*Alexandre, Nobre & Casteleiro, 2008*) | Polyp2007 (*Alexandre, Nobre & Casteleiro, 2008*) |

classifier resulted in a good detection scores (*Tajbakhsh et al., 2014*; *Pogorelov et al., 2017a*; *Iakovidis, Maroulis & Karkanis, 2006*; *Alexandre, Nobre & Casteleiro, 2008*; *Karkanis et al., 2001*; *Zhao et al., 2021*; *Bernal, Sánchez & Vilarino, 2013*; *Esgiar et al., 1998*; *Deeba et al., 2018*; *Hwang et al., 2007*). The F1-scores of the approaches of *Pogorelov et al. (2017a)*, *Zhao et al. (2021)*, *Hwang et al. (2007)* and *Deeba et al. (2018)* are above 0.90 for the detection of various abnormalities in the GI tract. Some approaches focus on the precision, recall, or the Matthews correlation coefficient (MCC) score, resulting in some of these evaluation parameters (*Tajbakhsh et al., 2014*; *Naqvi et al., 2017*). Table 1 shows the best evaluations on each of these parameters.

Several contributions are used to detect and classify abnormalities on some benchmark datasets. Some of the approaches are applied on the Kvasir dataset (*Pogorelov et al., 2017c*) with eight classes and an equal number of samples for all these classes. The approach of SCL-UMD (*Agrawal et al., 2017*) is one of the approaches applying two different deep learning models with handcrafted features. The approach yielded an excellent detection accuracy of 0.96 with an F1-score of 0.85. The deep learning algorithms of the VGG (*Szegedy et al., 2015*) and inception (*Szegedy et al., 2016*) are time taking approaches that reduces the speed of the SCL-UMD to 1.3 frames per second (FPS) (*Jha et al., 2021a*). The FPS rate of the detection is achieved better by the SIMULA-17 (*Pogorelov et al., 2017d*) by applying a Logistic model tree classifier on the handcrafted features. The approach resulted in a slightly lower accuracy with a detection speed of 46 FPS. The F1-score of the approach is 0.78 when applied to the dataset of Kvasir (*Pogorelov et al., 2017c*). With the recent success of deep learning, HCMUS (*Hoang et al., 2018*) applied the Faster-RCNN model for the abnormalities detection when the dataset has a variable number of samples for each class in the Kvasir V2 (*Pogorelov et al., 2018*). The approach improved the F1-score to 0.93 with the 23 FPS speed. The approach of FAST-NUDS based on only texture features resulted in the F1-score of 0.75 with a real-time detection (*Khan & Tahir, 2018*). The best evaluations on detecting the various abnormalities in the GI tract are achieved by *Zhao et al. (2021)* using a fusion of the features extracted from the deep neural network with the handcrafted features.

Classifying polyps, lesions, and other abnormalities within the GI tract is imperative for accurate diagnosis and subsequent treatment. This classification requires precise information about the specific regions of these abnormalities. The abnormalities can be a polyp, ulcer, lesion, esophagitis, or cancer. These abnormalities can be in various regions of the GI tract, *e.g.*, esophagitis can only be in the esophagus and before the z-line; however, the ulcer and polyps can be in various regions of the GI tract. The treatment needs regions of various abnormalities; for that, some work is available to detect abnormalities with anatomical landmarks. Some of the contributions to the detection of abnormalities with their classification and detection of the anatomical landmarks are using deep neural network architectures (*Krizhevsky, Sutskever & Hinton, 2012*; *He et al., 2016*; *Zeiler & Fergus, 2014*; *Simonyan & Zisserman, 2014*; *Szegedy et al., 2015*; *Huang et al., 2017*). *Chang et al. (2019)*, *Hoang et al. (2019)*, *Hoang et al. (2018)* and *Luo et al. (2019)* applied the neural network architectures with the attention and adaptive threshold to detect the abnormalities and the landmarks in the GI tract. The approaches resulted in an F1-score of more than 0.91. The analysis of various approaches is shown in Table 2.

There are some approaches for detecting the abnormalities and the landmarks with 23 classes in total, including various severity levels of the abnormalities. The approaches of the DDANet (*Tomar et al., 2021*), NanoNet (*Jha et al., 2021b*), and Data Bagging (*Khan et al., 2021*) resulted in good detection of F1-score of 0.83, 0.72 and 0.60, respectively. Some of the approaches are performing best in the case of real-time detection and classification. The Tiny Darknet (*Dutta, Bhattacharjee & Barbhuiya, 2021*) approach is best for the classification of abnormalities and landmark detection in real-time.

## PROPOSED APPROACH

In this section, the primary methodology of the research is discussed. The system consists of seven main components. Each component is explored *via* state-of-the-art techniques and discussed below.

1. Data preprocessing.
2. Data augmentation.
3. Handcrafted feature extraction.
4. Deep feature extraction.
5. Feature combination and selection.
6. Network-based classification.
7. Genetic algorithm for threshold detection.

### Data preprocessing

Reflections on the endoscopic images may arise during video capturing. It is necessary to remove reflections since the reflection of light may cause an erroneous image analysis, which can subsequently affect the abnormality and anatomical landmark detection. This reflection can be seen in the sample set of polyps picture in Fig. 1. Machine learning and computer vision (CV) based approaches work primarily based on color difference. The difference between polyp and dyed lifted polyp differs mainly based on color difference. A reflection can convert the whole red or blue color into a white color, which is not

**Table 2 Analysis of several various approaches for classification of abnormalities and landmarks detection.**

| Sr. No. | Approach | Acc. | F1-score | FPS | Dataset description |
|---|---|---|---|---|---|
| 1 | Adaptive Ensembles (*Luo et al., 2019*; *Jha et al., 2021a*) | 0.99 | 0.95 | 10 | Kvasir V2 dataset (*Pogorelov et al., 2018*) used with total images of (5,293 train and 8,737 test) 10,030 |
| 2 | FAST RCNN (*Hoang et al., 2018*; *Jha et al., 2021a*) | 0.99 | 0.94 | 23 | Kvasir V2 (*Pogorelov et al., 2018*) |
| 3 | Mobile Net (*Harzig, Einfalt & Lienhart, 2019*) | 0.99 | 0.88 | 3,226 | A combination of Kvasir V1 (*Pogorelov et al., 2017c*) and Kvasir V2 (*Pogorelov et al., 2018*) with the total images of (11,969 train and 8,740 test) 20,709 |
| 4 | Data Enhancement and ResNet50 (*Meng et al., 2019*) | 0.98 | 0.87 | 98 | A combination of Kvasir V2 (*Pogorelov et al., 2018*) and Nerthus (*Pogorelov et al., 2017b*) with the total images of (7,032 train and 1,758 test) 8,790 |
| 5 | DenseNet and Alexnet (*Hicks et al., 2018*) | 0.99 | 0.89 | 1,015 | Kvasir V2 dataset (*Pogorelov et al., 2018*) |
| 6 | Multi-Class Classifier Neural Network (*Hoang et al., 2019*) | 0.99 | 0.88 | 3.6 | Kvasir V2 dataset (*Pogorelov et al., 2018*) |
| 7 | Majority Voting (*Khan & Tahir, 2018*; *Jha et al., 2021a*) | 0.98 | 0.76 | 43,328 | Kvasir V2 dataset (*Pogorelov et al., 2018*) |
| 8 | Inception Res-Net (*Kirkerød et al., 2018*) | 0.99 | 0.92 | 20 | Kvasir V2 dataset (*Pogorelov et al., 2018*) |
| 9 | Weighted Discriminant Embedding (*Ko, Gu & Liu, 2018*; *Jha et al., 2021a*) | 0.95 | 0.48 | 3,744 | Kvasir V2 dataset (*Pogorelov et al., 2018*) |
| 10 | Hyper Parameter optimised DenseNet 169 (*García-Aguirre et al., 2022*) | 0.98 | 0.94 | 10 | Kvasir V1 dataset (*Pogorelov et al., 2017c*) |

distinguishable from the CV-based approaches. Some of the image samples were not correctly classified without preprocessing; however, they were classified after applying the reflection removal techniques. A few such image samples from the dataset of Kvasir V2 (*Pogorelov et al., 2018*) are shown in Fig. 1.

There are various procedures for removing these reflections with or without human intervention (*Criminisi, Pérez & Toyama, 2004*). Some of the standard reflection detection and removal procedures used in the research are as follows:

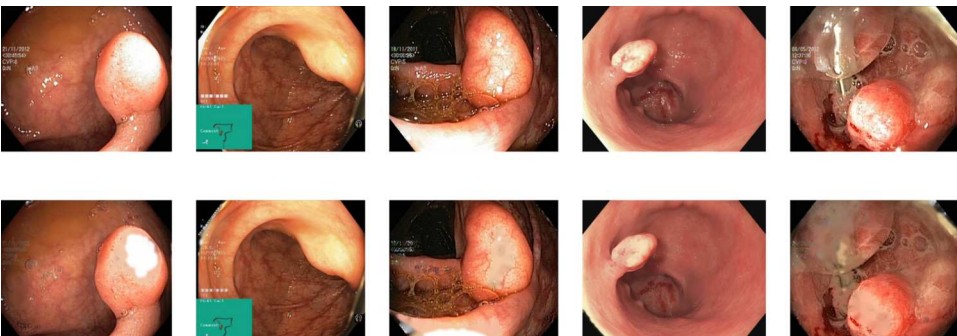

**Figure 1** **Reflections on the images of the Kvasir V2 (*Pogorelov et al., 2018*) polyps category.**

**Image crop:** The most straightforward way of reflection removal is to crop the image to remove the region of the reflection. This crop needs to compute the reflection region. This method reduces the image size, affecting the actual image in terms of the location of the abnormality. The cropping of the image can eliminate the region necessary for the abnormality detection. The method is suitable when the reflections only affect the image's background. The method is applied on various datasets of Kvasir and resulted in the average F1-score of 0.72 using various features and a decision tree classifier on the Kvasir V2 dataset (*Pogorelov et al., 2018*). After applying the image crop, the other datasets resulted in lower accuracy than the original images.

**Supervised detection:** Supervised learning-based reflection detection and removal is the reflection technique that resulted in good detection accuracy when applied to some of the images in a few datasets. The technique requires a massive amount of humanly annotated data for the training. A more minor data is humanly annotated as reflection removed. The small amount of data resulted in negligible accuracy improvements when used in benchmark datasets.

**Unsupervised detection:** The unsupervised learning approach for detecting the reflection detects the color layout difference at some of the image chunks that show some light reflection. The detection is based on the idea that a pixel with a reflection will be brighter. Experiments show that a brightness of more than 180 can be a light reflection grayscale image of endoscopy. In the usual frame, a pixel value can have a channel value but not all three channels. A pixel with values above 180 is considered the reflection and must be removed. The detection can be written as Eq. (1).

$$I_{x,y} \leftarrow \begin{cases} 255, & \text{if } I_{x,y} > 180 \\ 0, & \text{otherwise} \end{cases} \tag{1}$$

The above simple approach can detect the areas of solid reflection where a strong reflection generates the grayscale value of 180. There is some weak reflection where the grayscale pixel value is 150 to 180. The weak reflections can be the reflection or just a lighter image pixel. An enhancement approach is applied to get the reflection region with low reflection

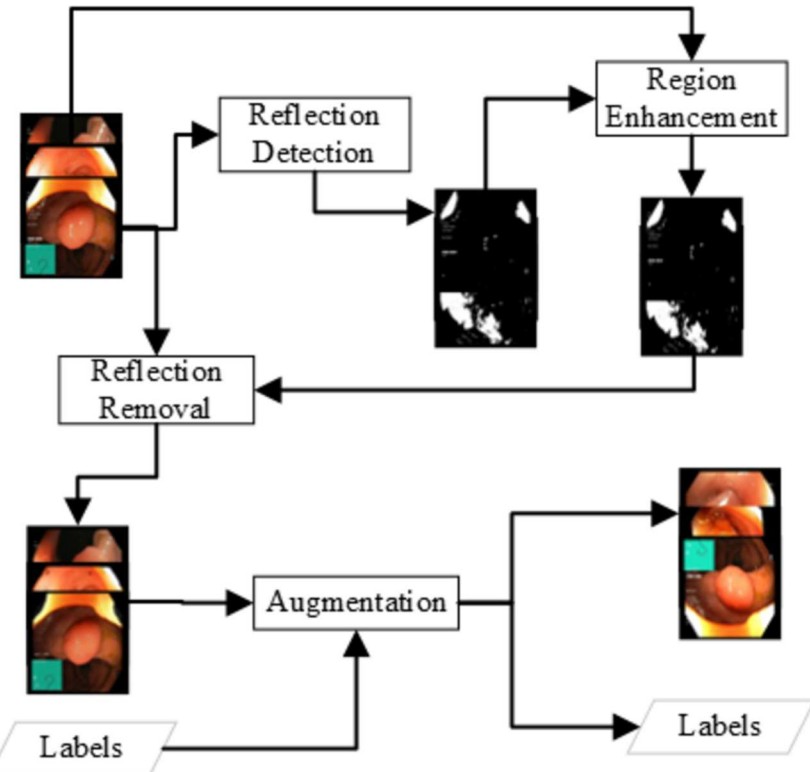

**Figure 2** Data preprocessing for the reflection removal and data augmentation for the class imbalance problem.

probabilities. The methodology for detecting weak reflections is based on the approach of expanding the reflection. All the pixels where the pixel is a weak reflection (150 to 180) and the adjacent pixel is a reflection, the pixel will be considered as reflection. The Eqs. (2), (3) can represent the reflection enhancement. The Reflection removal procedure is shown in Fig. 2.

$$I_{x,y} \leftarrow \begin{cases} 255, & \text{if } I_{x,y} > 130 \wedge \exists Adj(I_{x,y}) > 180 \\ 0, & \text{otherwise} \end{cases} \tag{2}$$

$$Adj(I_{x,y}) \leftarrow I_{x-i,y-j} \forall (i \in 1,0,-1) \forall (j \in 1,0,-1) \tag{3}$$

The unsupervised removal approaches are available in openCV in the paint with the Fast Marching Method by *Telea (2004)* or the idea of the Navier–Stokes fluid dynamics (*Bertalmio, Bertozzi & Sapiro, 2001*). In this research, this unsupervised detection and removal method of Telea resulted in the best detection in terms of accuracy detection and speed. The method resulted in an average F1-score of 0.84 using various features and a decision tree classifier, which is better than Image Crop.

## Data augmentation

A few datasets are available for endoscopic procedures, especially for some of the abnormalities. Several abnormalities are found in most patients, while some are rare in very few patients. There exists a high-class imbalance in some of the benchmark datasets. The dataset of Kvasir V2 has a minor class with just four samples and a major class with 613 image samples (*Pogorelov et al., 2018*). Similarly, in version 3, named Hyper Kvasir, this difference is more than the previous one with the minor class of hemorrhoids with six samples and major class of BBPS 2–3 with 1,148 samples (*Borgli et al., 2020*). Various ML and DL approaches, when applied to the class imbalance datasets of the Kvasir, resulted in good detection for the major class and worst for the minor class (*Hoang et al., 2018*; *Dias & Dias, 2018*; *Khan & Tahir, 2018*). Another limitation in the endoscopic datasets is the limited availability of datasets. The limitation of the data and the class imbalance can be mitigated by data augmentation.

The common approaches for data augmentation are augmentation using generative models (GANs) (*Goodfellow et al., 2014*) or auto-encoder methods and data augmentation by image manipulation. The GANs-based data augmentation resulted in some problems regarding the decision. The negative aspect of the GANs-based augmentation is the time required by the GANs to augment images. The approach of image manipulation is time efficient with different accuracies on different datasets in many cases in comparison to GANs (*Shijie et al., 2017*; *Shorten & Khoshgoftaar, 2019*; *Ethiraj & Bolla, 2022*; *Kamruzzaman et al., 2022*). In this research, both approaches were applied on the Kvasir V2 dataset (*Pogorelov et al., 2018*) for the comparison, which resulted in better accuracy and training time with the image manipulation methods. The manipulations used in the research are as follows:

- Rotate (Random Angles)
- Flip (Random Angles)
- Crop (Random size from original to $256 \times 256$)
- Resize (Random size from original to $256 \times 256$)
- Noise injection (*Tammina, 2022*)

The image manipulation is done as a post-step of the reflection removal and a pre-step to the feature extraction and fine-tuning of the neural network as shown in Fig. 2. The images generated in such a way that all the classes should have similar amount of image samples. The number of images generated for each image I from class C ($C \in Classes$) are as shown in the Eq. (4).

$$Generate\{I, C, Classes\} \leftarrow \lfloor \{max_{Ci \in Classes}(count(Ci)) * 1.1 - count(C)\} \rfloor \qquad (4)$$

where count(C) is the number of images available in the class C from which class the current image belongs to and count (Ci) is the number of images available in class Ci.

The average detection F1-score of the image manipulation method is above 0.84 by applying the decision tree classifier on various sets of features.

## Texture feature extraction

There are several state-of-the-art features shown as good deciders. Some of the texture features are available in Lucene Image Retrieval (LIRE) (*Lux & Chatzichristofis, 2008*),

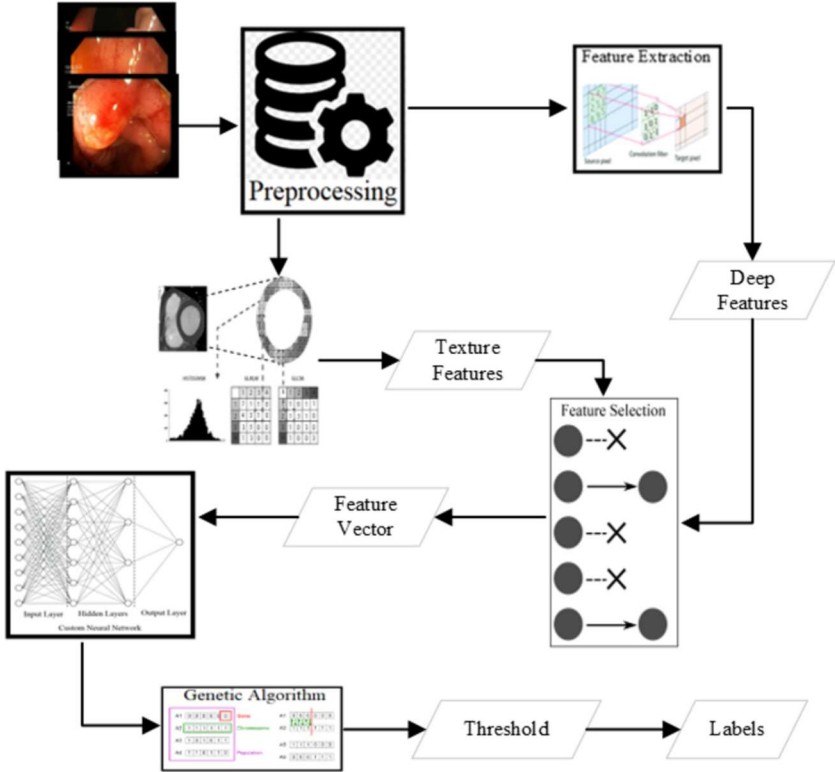

**Figure 3   Methodology of the research.**

Local binary and ternary patterns (*Ojala, Pietikäinen & Harwood, 1996*), and gray level co-variance matrix statistics named as Haralick features (*Haralick & Shanmugam, 1973*) are extracted to use for the classification. The state-of-the-art features are an excellent decider for some of the classes. Various combinations of these features have been applied for the classification using the voting classifier shown in Fig. 3. The best results are produced from some of the LIRE features (*Lux & Chatzichristofis, 2008*) with Gray-Level Co-Occurrence Matrix (GLCM) features (*Haralick & Shanmugam, 1973*; *Ojala, Pietikäinen & Harwood, 1996*).

The most common image features used for the detection of abnormalities and their localization and segmentation are as follows:

**Color Layout:** The color layout is the spatial distribution of the color of a region of image (*Sikora, 2001*). It is a resolution-invariant image color representation as a discriminator irrespective of resolution.

**Edge Histogram:** Edge histogram is a non-homogeneous texture descriptor to capture the spatial distribution of the edges (*Sikora, 2001*). The edges are the color change, so the edge histogram is very close to the color layout descriptor in decision-making.

**Tamura:** Tamura computed the six image texture features: coarse, contrast, direction, line likeness, regularity, and roughness (*Tamura, Mori & Yamawaki, 1978*). These six features

are combined with the Tamura features based on the author's name. The combination of these features is a good representation of an image.

**Color and Edge Directivity Descriptor (CEDD):** CEDD is a feature of the image that has the information of the color and texture in a histogram (*Chatzichristofis & Boutalis, 2008a*). The computation of these features is faster than many other texture and deep features.

**Fuzzy color and texture histogram (FCTH):** FCTH is a low-level representation of the texture and color histogram (*Chatzichristofis & Boutalis, 2008b*).

**Color histograms (HSV and RGB):** The color histogram is the representation of the color distribution in the image (*Huang et al., 1997*; *Lux & Chatzichristofis, 2008*; *Lux, 2013*). The Color histogram can be computed as shown in Eq. (5).

$$H(I,C) = \forall_{p \in I, c \in C} P(p_x == c_i) \tag{5}$$

**Gabor Features:** The Gabor filters are the product of sinusoidal wave and Gaussian function (*Mehrotra, Namuduri & Ranganathan, 1992*; *Lux & Chatzichristofis, 2008*). The filter represents the edges in an image for better detection of any object inside an image.

**Auto Color Correlation:** The Auto Color Correlation is a relationship matrix between colors of various pixels of the image (*Lux & Chatzichristofis, 2008*; *Lux, 2013*)

**Auto Color Correlogram:** *Huang et al. (1997)* proposed a feature of the image to define the color distribution of the image. The feature has information on the probability of a color difference of pixels in the image. The correlogram can be computed for the Image $I$ with pixels $p_{1,2,3,...,n}$ and the colors $C = C_{1,2,3,...,m}$ and the distance of $k$ as Eq. (6).

$$ACC(I,C,k) = P(p_x == c_i \wedge p_y == c_j \wedge d(p_x, p_y) = k) \tag{6}$$

$$d(P,Q) = max(|P_x - Q_x|, |P_y - Q_y|) \tag{7}$$

The distance used in the correlogram is the difference of maximum pixel as shown in Eq. (7).

**Speeded up robust features (SURF):** SURF is a scale-invariant and rotation-invariant descriptor of image (*Bay, Tuytelaars & Gool, 2006*). These features are faster in extraction and usable because of scale and rotation independence.

**Pyramid Histogram of Oriented Gradients (PHOG):** PHOG are the grids of Histograms of Oriented Gradient descriptors (*Dalal & Triggs, 2005*). The features are the best descriptors for human identification and perform well in the tasks of endoscopic image classification.

**Local Binary Patterns (LBP):** Local Binary Patterns (LBP) is a texture representation of an image or section of the image that contains the gray level relation of a pixel with its neighbors (*Ojala, Pietikäinen & Harwood, 1996*; *Liu et al., 2016*; *Liao, Law & Chung, 2009*; *Zhu, Bichot & Chen, 2010*). A circular sum of the values of the neighborhood of each pixel calculates the LPBs. The radius parameter defines the neighborhood. Every pixel in the image is used as a center pixel, and the neighboring pixels with radius r are assigned a value of 0 if the value of a neighboring pixel is lesser than the value at the center and 1 otherwise. The number made by combining bits of the circle, starting from the top middle, is the LBP

value for the central pixel. The LBP of an image is a matrix of the same size as the image. The LBP can be computed as shown in the Eqs. (9) and (8).

$$P(i) \leftarrow \begin{cases} 1, & \text{if } X(x,y) \geq I(i,j) \\ 0, & \text{otherwise} \end{cases} \tag{8}$$

$$LBP(i,j) \leftarrow \sum_{\forall p \in P} 2^p \tag{9}$$

**Local Ternary Patterns (LTP):** Similar to the local binary patterns there is another set of patterns that uses three states instead of just binary known as Local Ternary Patterns (*Tan & Triggs, 2010*). The computation of the LTP is shown in Eq. (10).

$$P(i) \leftarrow \begin{cases} +1, & \text{if } X(x,y) > I(i,j) \\ -1, & \text{if } X(x,y) < I(i,j) \\ 0, & \text{otherwise} \end{cases} \tag{10}$$

The rest of the calculation of the LTP is similar to the LBP.

**Gray-Level Co-Occurrence Matrix (GLCM):** The CLCM contains information on the number of concurrence of the different color contrasts in the horizontal, vertical, upper diagonal, and lower diagonal neighboring (*Haralick & Shanmugam, 1973*). The neighbors of a pixel $P$ are pixels that are at distance $d$ from $P$. The horizontal GLCM can be computed as in Eq. (11).

$$GLCM(x,y,d,hr) \leftarrow count(\forall_{i,j,k}(I_{i,j} = x \wedge I_{i,k} = y \wedge |j-k| = d)) \tag{11}$$

The $hr$ is a representation of the direction that is horizontal or 0 degrees. Similar computations for other angles or directions can also be computed by just a change in some of the pixels in the equation. The vertical GLCM computation is as shown in Eq. (12).

$$GLCM(x,y,d,vr) \leftarrow count(\forall_{i,j,k}(I_{i,j} = x \wedge I_{k,j} = y \wedge |i-k| = d)) \tag{12}$$

**Haralick Features (Statistics of GLCM):** Haralick proposed some texture features computed as a set of 14 statistics on the GLCM matrices (*Haralick & Shanmugam, 1973*). One of those statistics is shown in Eq. (13).

$$Haralick(I) = \sum_{i=1}^{n} \sum_{j=1}^{n} (P(i,j)/R) \tag{13}$$

The value $P(i,j)$ is the pixel value at $row = i$ and $column = j$ and $R$ is a normalization constant.

### Deep feature extraction

Various CNN-based approaches are used to detect abnormality in endoscopy and colonoscopy images. The best detection accuracies have been achieved using Dense

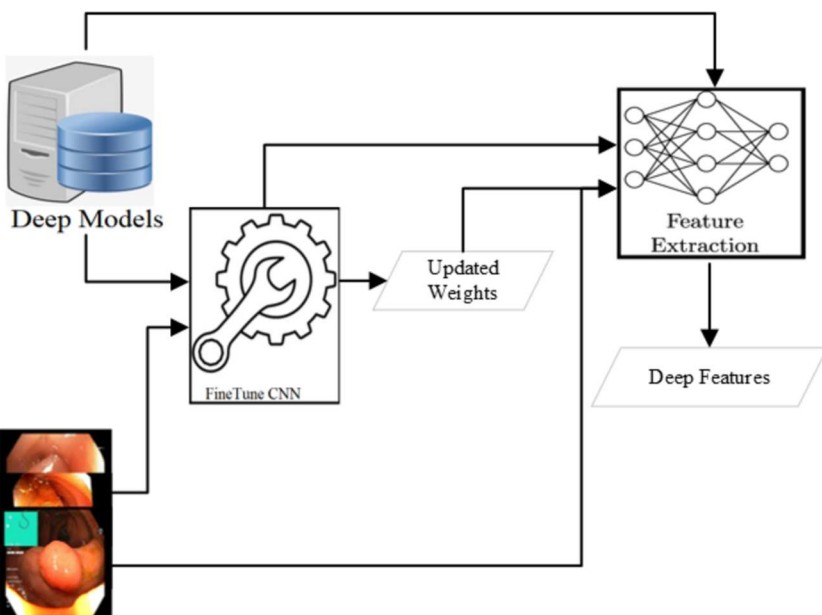

**Figure 4** Deep feature extraction methodology.

Net (*Huang et al., 2017*; *Hicks et al., 2018*; *Jha et al., 2021a*), Res Net (*He et al., 2016*; *Hoang et al., 2018*; *Kirkerød et al., 2018*; *Jha et al., 2021a*) and Mobile Net V2 (*Howard et al., 2017*; *Sandler et al., 2018*; *Harzig, Einfalt & Lienhart, 2019*; *Jha et al., 2021a*). The detection of all these three approaches is comparable after applying various preprocessing steps. However, the Mobile Net v2 detection speed is the highest among these three. In this research, the Mobile Net v2 (*Howard et al., 2017*; *Sandler et al., 2018*) is fine-tuned to use as a deep feature extractor as shown in Fig. 4. The Mobile Net V2 is a lighter network with 105 layers. The inference time of the network is around 41 images per second when executed on a CPU machine while more than six times on a GPU machine. The network's last layer resulted in the 1,000 class output with the probabilities of the objects when trained on the Image Net dataset (*Deng et al., 2009*). After the Flatten operation, the output obtained from the first layer yields a deep feature characterized by a feature size of 1,280.

## Feature selection

The approach of the random combination of various features for the classification using a voting classifier resulted in good detection with optimal classification time. The final set of features resulted in the best accuracies are as follows:

- Auto Color Correlogram (*Lux & Chatzichristofis, 2008*)
- Color Layout (*Lux & Chatzichristofis, 2008*; *Sikora, 2001*; *Huang et al., 1997*)
- Edge Histogram (*Lux & Chatzichristofis, 2008*; *Sikora, 2001*)
- Gabor Features (*Lux & Chatzichristofis, 2008*)
- JCD (Joint Collection Distances) (*Lux & Chatzichristofis, 2008*)

- Color and Edge Directivity Descriptor (*Lux & Chatzichristofis, 2008*; *Chatzichristofis & Boutalis, 2008a*)
- Fuzzy Color and Texture Histogram (*Lux & Chatzichristofis, 2008*; *Chatzichristofis & Boutalis, 2008b*)
- Tamura Features (Contrast, Direction *etc.*) (*Lux & Chatzichristofis, 2008*; *Tamura, Mori & Yamawaki, 1978*)
- Local Binary Patterns (Radius from 1 to 5) (*Haralick & Shanmugam, 1973*; *Ojala, Pietikäinen & Harwood, 1996*)
- Deep Features (*Howard et al., 2017*; *Sandler et al., 2018*; *Harzig, Einfalt & Lienhart, 2019*)

## Classification network

Neural Networks are good deciders on the availability of sufficient data (*Deng et al., 2009*; *Simonyan & Zisserman, 2014*). A classification neural network can be divided into the feature extractor and the decider. In the previous steps of the approach, deep learning is used as a feature extractor. In this step, the decider component of the neural network is applied to the selected set of feature vectors with values ranging from 0 to 1. The decider neural network used in the research consists of three layers based on a balance between generalization, detection accuracy, and detection time. To include non-linearity, for better generalization, some activation functions were evaluated. The feature vector has values ranging from 0 to 1, which resulted in much smaller activations in our application of the fully connected layer. The most suitable activation function for adding non-linearity and removing gradient descent was Relu (*Agarap, 2018*) as shown in the Eq. (14). The same Rule function is used in the second layer output for the non-linearity.

$$Relu(x) = max(0, x) \tag{14}$$

The final output from the third layer was needed to be the probability of the various classes for each of the input feature vectors. A sigmoid activation function, shown in Eq. (15), is used at the end of the last layer for converting results from any range to the range of zero to one.

$$f(x) = \sigma(x) = \frac{1}{1 + e^{-x}} \tag{15}$$

The complete forward path of the neural network can be represented by the Eq. (16).

$$Y = \sigma(W_3^T * Relu(W_2^T * Relu(W_1^T * X + B_1) + B_2) + B_3) \tag{16}$$

Various optimizers were evaluated for the neural network. Based on the good weights adjustment and fast convergence, Nadam (shown in Eq. (17)) optimiser is used in this research (*Dozat, 2016*).

$$Y_{i+1} = Y_i - \frac{\eta}{\sqrt{\hat{v}_t} + \epsilon} \left( \beta_1 \hat{m}_t + \frac{(1 - \beta_1) g_t}{1 - \beta_1^t} + \frac{\beta_1 \beta_2}{1 - \beta_1^t} m_t \right) \tag{17}$$

where:

- $i$ represents the current iteration.

- $i+1$ represents the next iteration.
- $m_t$ is the first moment (exponential moving average of the gradient).
- $v_t$ is the second moment (exponential moving average of the squared gradient).
- $g_t$ is the current gradient.
- $\hat{m}_t$ and $\hat{v}_t$ are bias-corrected versions of $m_t$ and $v_t$.
- $i+1$ is the updated parameter value.
- $\eta$ is the learning rate.
- $\beta_1$ and $\beta_2$ are the exponential decay rates for the moment estimates.
- $\epsilon$ is a small constant added for numerical stability.

The accuracy of this approach is 0.99 with the F1-score of 0.89 and Matthews Correlation Coefficient (MCC) (*Guilford, 1954*; *Matthews, 1975*) of 0.88.

The configurations of the network used are as follows:

1. Layer 1: 64 Fully Connected Neurons with Relu Activation Function
2. Layer 2: 64 Fully Connected Neurons with Relu Activation Function
3. Layer 3: 64 Fully Connected Neurons with Sigmoid Activation Function

The NN-based approach resulted in a boost in F1-score from 0.89 to 0.90 and the Matthews Correlation Coefficient (MCC) (*Guilford, 1954*; *Matthews, 1975*) from 0.88 to 0.90 in the same detection time of 41 frames per second (FPS).

## Genetic algorithm for threshold detection

The intra-class and inter-class differences are different in different classes. For example, the difference between polyp and z-line class is much higher than between z-line and esophagitis. The inter-class difference resulted in the low detection probabilities, detected by a neural network architecture for some classes, and higher for others.

The detection is boosted with threshold detection using a genetic algorithm (GA-Boost). The GA-Boost algorithm learned the thresholds for each detected class by applying the genetic algorithm starting with a random threshold from these ten $(0.0, 0.1, 0.2, 0.3, 0.4, 0.5, 0.6, 0.7, 0.8, 0.9)$. The crossover operator employed in GA-Boost is an addition with modulus operation. It involves removing the integral part of the floating-point number to ensure that the resulting number lies within the range of 0 to 1. Mathematically, the operator can be represented as Eq. (18), where $x$ and $y$ denote the two original chromosomes, and $X_n \in (0, 1]$ and $Y_n \in (0, 1]$ represent the two newly generated chromosomes. The GA algorithm used the population size of 10 chromosomes generated from random values of 0 to 1. The mutation size is 20% and it is executed for the 20 iterations to get the best results. The evaluation measure used in the crossover selection is the F1-score of abnormalities detection using the Decision Tree on the combined feature vector of the Kvasir V2 dataset (*Pogorelov et al., 2018*). The GA-Boost boosted the accuracy measures to an F1-score of 0.91.

$$(X_n, Y_n) \leftarrow ((x+2y) \bmod 1, (2x+y) \bmod 1) \tag{18}$$

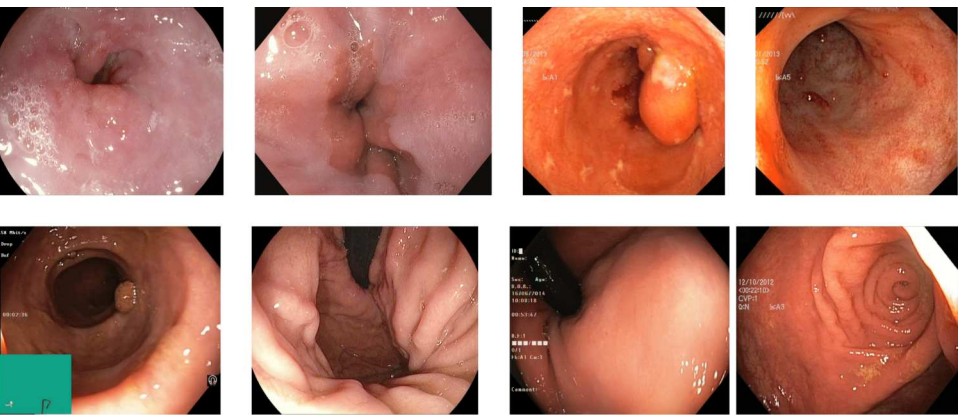

**Figure 5** Some images of various classes from Kvasir V2 (*Pogorelov et al., 2018*).

# EXPERIMENTAL SETUP

This section discusses the experimental setting used for the empirical analysis of the proposed methodology. It includes a description of the datasets, details of the evaluation measures, and the comparison methods used in the research.

## Datasets

There are several datasets available for the analysis of endoscopic or colonoscopic abnormalities. Some datasets are annotated for detecting polyps or lesions in the endoscopy. This research focuses on the detection of the abnormalities, so the datasets for the detection are listed below:

**Kvasir V1** The Kvasir V1 dataset comprises 8,000 images with 4,000 training and 4,000 testing images. The images are of various sizes from $720 \times 576$ to $1,920 \times 1,072$ pixels. There are eight classes of GI tract, including some anatomical landmarks, pathological and normal findings, or endoscopic procedures (*Pogorelov et al., 2017c*). Each class has the same number of image samples in the dataset. The list of classes is as follows:

- dyed-lifted-polyps
- dyed-resection-margins
- esophagitis
- normal-cecum
- normal-pylorus
- normal-z-line
- polyps
- ulcerative-colitis

**Kvasir V2:** The Kvasir V2 dataset comprises 14,033 images with 5,293 training and 8,741 testing images. The images are of various sizes from $720 \times 576$ to $1,920 \times 1,072$ pixels. There are 16 classes of GI tract, including some anatomical landmarks, pathological and normal findings, or endoscopic procedures (*Pogorelov et al., 2018*). Some pictures of various abnormalities and landmarks are shown in Fig. 5. The dataset has a high-class

**Table 3 Class distribution in various datasets.**

| Sr. No. | Class name | Kvasir V1 (*Pogorelov et al., 2017c*) | Kvasir V2 (*Pogorelov et al., 2018*) | Hyper Kvasir (*Borgli et al., 2020*) | DowPK (*Khan, 2023*) |
|---|---|---|---|---|---|
| 1 | barretts | – | – | 41 | – |
| 2 | barretts short segment | – | – | 53 | – |
| 3 | BBPS 0–1 | – | – | 646 | – |
| 4 | BBPS 2–3 | – | – | 1,148 | – |
| 5 | blurry nothing | – | 213 | – | 114 |
| 6 | colon clear | – | 1,332 | – | 38 |
| 7 | dyed lifted polyps | 500 | 1,013 | 1,002 | 7 |
| 8 | dyed resection margins | 500 | 980 | 989 | 85 |
| 9 | esophagitis A | 500 | 1,000 | 403 | 28 |
| 10 | esophagitis B-D | – | – | 260 | – |
| 11 | hemmorrhoids | – | – | 6 | – |
| 12 | impacted stools | – | 636 | 131 | 6 |
| 13 | instruments | – | 309 | – | 20 |
| 14 | normal cecum | 500 | 1,000 | 1,009 | 16 |
| 15 | normal pylorus | 500 | 1,000 | 999 | 35 |
| 16 | normal z-line | 500 | 1,000 | 932 | 135 |
| 17 | out of patient | – | 9 | – | 23 |
| 18 | polyps | 500 | 987 | 1,028 | 104 |
| 19 | retroflex rectum | – | 429 | 391 | 82 |
| 20 | retroflex stomach | – | 795 | 764 | 42 |
| 21 | stool plenty | – | 2,331 | – | 36 |
| 22 | Terminal Illunium | – | – | 9 | – |
| 23 | ulcerative colitis grade 0–1 | 500 | 999 | 35 | 70 |
| 24 | ulcerative colitis grade 1 | – | – | 201 | – |
| 25 | ulcerative colitis grade 1–2 | – | – | 11 | – |
| 26 | ulcerative colitis grade 2 | – | – | 443 | – |
| 27 | ulcerative colitis grade 2–3 | – | – | 28 | – |
| 28 | ulcerative colitis grade 3 | – | – | 133 | – |

imbalance. The classes with the number of image samples are listed below in Table 3. Some classes have just nine samples with four training and five test samples, while others have 2,331 samples.

**Hyper Kvasir Classification:** The Hyper Kvasir Classification dataset consists of 110,079 images and 374 videos. The dataset used in the challenge is a part of this data with 10,662 training images of 23 classes and 721 new labeled images for the anatomical landmarks, pathological and normal findings, or endoscopic procedures (*Borgli et al., 2020*). The class distribution is shown in Table 3.

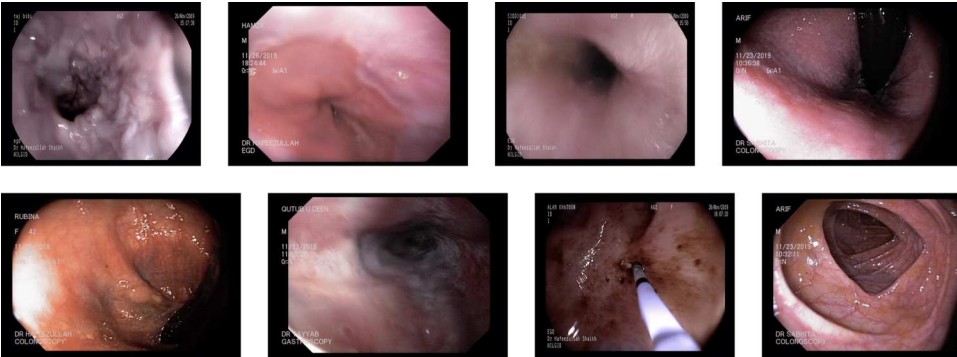

**Figure 6** Some images of various classes from DowPK dataset (*Khan, 2023*).

**DowPK:** The dataset, retrieved from Dow Hospital Pakistan, consists of 841 images of various abnormalities in the endoscopy and colonoscopy. The images are of various patients, with an average of 4 images from each patient. These images were annotated by a group of 8 MBBS final year students and computed the majority voting. Most of the images got the same class from all the annotators. The images that have different classes by the annotations done by the different medical experts were decided based on majority voting. Some of the images from the DowPK dataset are shown in Fig. 6. The class distribution of the dataset is shown in the Table 3.

## Evaluation measures

Various evaluation measures have been used for the evaluation of the segmentation. Some of the evaluation measures used for the research are as follows:

**Accuracy:** Accuracy is the number of true classes over the total number of classes in the task. The accuracy of the methodology on various datasets was 0.99.

**F1-score:** The F1-score measures accurateness, a function of precision and recall. The F1-score achieved by the proposed approach for the Kvasir v2 dataset is 0.91, while 0.93 on the DowPK dataset.

**Matthews Correlation Coefficient (MCC):** Various classification accuracy measures have their limitations. The precision, recall, and F1-score are asymmetric measures while the accuracy is sensitive to class imbalance. MCC is a difference between true and false prediction with the ratio of true prediction with all vales (*Guilford, 1954*; *Matthews, 1975*).

**Sensitivity:** Sensitivity, also known as the actual positive rate or recall, measures the proportion of correctly identified positive instances out of all actual positive instances. It is a metric used to evaluate the ability of a model to classify positive instances correctly. A high sensitivity value indicates that the model effectively identifies positive instances and minimizes false negatives.

The DowPK dataset demonstrated the highest sensitivity value among the evaluated datasets, reaching an impressive value of 0.93. Sensitivity, also known as the true positive rate or recall, measures the proportion of correctly identified positive instances out of all actual positive instances. A sensitivity value of 0.93 indicates that the model achieved a

high level of accuracy in correctly identifying the positive instances within the DowPK dataset. This highlights the model's strong performance in detecting the desired outcomes or conditions specific to the DowPK dataset.

**Specificity:** Specificity refers to the proportion of correctly identified negative instances out of all actual negative instances. It is a metric used to evaluate the ability of a model to classify negative instances correctly. A high specificity value indicates that the model effectively distinguishes negative instances and minimizes false positives.

The DowPK dataset showcased the highest specificity value of 0.93 among the assessed datasets. Specificity gauges the ratio of accurately identified negative instances among all the negatives. A specificity score of 0.93 underscores the model's proficiency in recognizing negative instances within the DowPK dataset.

**Area under the receiver operating characteristic curve (AUC-ROC):** The area under the receiver operating characteristic curve (AUC-ROC) is a widely used metric that assesses a model's ability to differentiate between positive and negative instances across various threshold settings (*McClish, 1989*). The one-vs-rest approach has been opted for this research which involves a multi-class dataset. The AUC-ROC concisely summarizes the model's discriminatory capabilities with higher values indicating superior performance in distinguishing between classes. It serves as a valuable metric for evaluating the model's overall effectiveness in capturing the dataset's inherent class separations.

The Kvasir V2 and Hyper Kvasir datasets demonstrated the highest AUC-ROC values among the evaluated datasets, both achieving an impressive value of 0.99. A value of 0.99 indicates that the model achieved excellent performance in distinguishing between positive and negative instances within these datasets. This high AUC-ROC value highlights the model's ability to classify instances accurately and effectively separate the two classes in the Kvasir V2 and Hyper Kvasir datasets.

When all the samples have been correctly classified, the value of the MCC will be one due to the ratio between the product of TP and TN over the same. The minimum possible value of the MCC can be -1 when all the results are false, and the values of FP and FN are maximum while the values of TP and TN are minimum.

### Compared methods

In this section, we will first discuss various variants of the proposed methods, followed by comparing those methods on benchmark datasets. A comparison of the various approaches is shown in Tables 4, 5 and 6.

## RESULTS AND DISCUSSION

This section presents the results of various steps in terms of improvements in detection accuracy and time. The section also discusses the impact of each step on the final detection results. Several possible options were used in the literature in each step and evaluated theoretically in 'Proposed Approach'. This section elaborates on the empirical impact of each possible option in all steps. The steps of the approach are listed below, with the impact on features, interim results, and the final detection results.

**Table 4** Analysis of some of the approaches for classification of abnormalities and landmarks detection on the Kvasir V1 dataset (*Pogorelov et al., 2017c*.

| Sr. No. | Approach | Acc. | F1-score | MCC | FPS |
|---------|----------|------|----------|-----|-----|
| 1 | HKBU17 (*Liu, Gu & Cheung, 2017*; *Jha et al., 2021a*) | 0.93 | 0.70 | 0.66 | 2 |
| 2 | Ensemble17 (*Naqvi et al., 2017*; *Jha et al., 2021a*) | 0.94 | 0.77 | 0.74 | 2 |
| 3 | Inception17 (*Petscharnig, Schöffmann & Lux, 2017*) | 0.94 | 0.76 | 0.72 | 1 |
| 4 | SCL-UMD17 (*Agrawal et al., 2017*) | 0.96 | 0.85 | 0.83 | 1 |
| 5 | DLGF17 (*Pogorelov et al., 2017d*; *Jha et al., 2021a*) | 0.96 | 0.83 | 0.79 | 46 |
| 6 | Lire-CNN (Proposed) | 0.99 | 0.90 | 0.90 | 41 |

**Table 5** Analysis of various approaches for classification of abnormalities and landmarks detection on the Kvasir V2 dataset (*Pogorelov et al., 2018*).

| Sr. No. | Approach | Acc. | F1-score | MCC | FPS |
|---------|----------|------|----------|-----|-----|
| 1 | EEIC19 (*Hoang et al., 2019*) | 0.99 | 0.95 | 0.94 | 23 |
| 1 | EEIC19 Fast (*Hoang et al., 2019*) | 0.99 | 0.88 | 0.89 | 3 |
| 2 | TLCSS18 (*Dias & Dias, 2018*; *Jha et al., 2021a*) | 0.98 | 0.87 | 0.89 | 3 |
| 2 | TLCSS18 Fast (*Dias & Dias, 2018*; *Jha et al., 2021a*) | 0.99 | 0.91 | 0.90 | 27 |
| 3 | FRCSS18 (*Hoang et al., 2018*; *Jha et al., 2021a*) | 0.99 | 0.94 | 0.94 | 23 |
| 4 | DSTL18 (*Hicks et al., 2018*; *Jha et al., 2021a*) | 0.99 | 0.89 | 0.89 | 1,015 |
| 5 | WDE18 (*Ko, Gu & Liu, 2018*) | 0.95 | 0.48 | 0.54 | 3,744 |
| 6 | MVHC18 (*Khan & Tahir, 2018*) | 0.98 | 0.75 | 0.81 | 43,328 |
| 7 | Lire-CNN (Proposed) | 0.99 | 0.90 | 0.90 | 41 |

**Table 6** Analysis of several approaches for classification of abnormalities and landmarks detection on the Kvasir V3 (Hyper Kvasir) dataset (*Borgli et al., 2020*).

| Sr. No. | Approach | Acc. | F1-score | MCC | FPS |
|---------|----------|------|----------|-----|-----|
| 1 | HMTA (*Galdran, Carneiro & Ballester, 2021*) | 0.99 | 0.87 | 0.86 | 20 |
| 2 | HLNN (*He et al., 2021*) | 0.98 | 0.90 | 0.89 | 129 |
| 3 | EDTDE (*Dutta, Bhattacharjee & Barbhuiya, 2021*) | 0.93 | 0.76 | 0.76 | 49 |
| 4 | Lire-CNN (Proposed) | 0.99 | 0.91 | 0.91 | 41 |

## Reflection removal

Both of the most common reflection removal approaches of the Image Crop and the Unsupervised Detection were applied to multiple datasets including Kvasir V1 (*Pogorelov et al., 2017c*), Kvasir V2 (*Pogorelov et al., 2018*), Kvasir V3 (*Borgli et al., 2020*) and DowPK. Various features were extracted from all three copies of the datasets, *i.e.,* the original dataset, reflection removed using image crop, and reflection removed using Telea. The decision tree classifier was used to get the F1-score from all three copies to get the best copy for further steps. The results varied for all the datasets, as shown in Table 7. The results of all Kvasir datasets were reduced by applying Image Crop from the original dataset.

Similarly, Telea reflection removal boosted the F1-score when applied to the Kvasir V1, Kvasir V2, and the dowPK datasets. The dataset of DowPK resulted in no improvements by applying Image Crop reflection removal and improved with the application of unsupervised

**Table 7   Impact of reflection removal on the abnormalities detection F1-score.**

| Dataset | Original | Telea ref. | Image crop |
|---|---|---|---|
| Kvasir V1 (*Pogorelov et al., 2017c*) | 0.70 | 0.71 | 0.52 |
| Kvasir V2 (*Pogorelov et al., 2018*) | 0.81 | 0.82 | 0.47 |
| Kvasir V3 (*Borgli et al., 2020*) | 0.82 | 0.82 | 0.61 |
| DowPK | 0.62 | 0.68 | 0.63 |

**Table 8   Impact of data augmentation on the detection F-1 score.**

| Dataset | Original | Augmentation |
|---|---|---|
| Kvasir V1 (*Pogorelov et al., 2017c*) | 0.71 | 0.71 |
| Kvasir V2 (*Pogorelov et al., 2018*) | 0.82 | 0.84 |
| Kvasir V3 (*Borgli et al., 2020*) | 0.82 | 0.83 |
| Dow | 0.68 | 0.79 |

reflection detection and removal using Telea. The comparison of the F1-score of the various datasets with Telea-based reflection removal, Image Crop, and without any reflection removal is shown in Table 7.

The impact of reflection removal varied across different datasets, and the results can be attributed to two key data characteristics. Firstly, the number of images within a class containing reflections influenced the effect of reflection removal. When a class had more images, the impact of reflection removal was comparatively lower, resulting in fewer improvements. Secondly, the amount of reflections in the dataset played a role. Datasets with a higher prevalence of reflections experienced a more significant impact from reflection removal techniques.

For instance, the dowPK and Kvasir V1 datasets contained a smaller number of images compared to the Kvasir V2 and Hyper Kvasir datasets. Consequently, as illustrated in Table 7, the contrast in F1-score between including and excluding reflection removal was notably more significant in the former datasets. Additionally, the DowPK dataset had a higher density of reflections, further contributing to the significant effect of reflection removal techniques on this dataset.

These insights provide an understanding of the variations in the impact of reflection removal across datasets and highlight the influence of factors such as the number of images per class and the amount of reflections present.

## Data augmentation

The data augmentation technique is applied to the class imbalance problem. The class imbalance was higher in versions 2 and 3 of Kvasir. The augmentation is applied on all the datasets, and it improved the detection significantly in versions 2 and 3 of the Kvasir and affected very low Kvasir V1 datasets where there was no class imbalance. The detection results of the augmented data and un-augmented data are shown in Table 8.

## Feature selection

The approach of deep features with texture-based features has been applied and it resulted in an accuracy of 0.99 with the F1-Score of 0.86 on the Kvasir V2 dataset (*Pogorelov et al., 2018*), which is an improvement compared to the usage of only deep or only texture features.

**Feature selection**: There were some misleading features per the individual feature accuracy. Some feature having low accuracy led to applying some feature selection methodologies. Various sets of randomly selected features were used for detection, and the best feature set consisting of some of the Lire features, deep features, and the Local binary patterns with a radius from 1 to 5 improved accuracy. A list of features for resulting in the best F1-score of 0.88 on the Kvasir V2 dataset (*Pogorelov et al., 2018*) is listed below:

- Auto Color Correlogram (*Huang et al., 1997*; *Lux & Chatzichristofis, 2008*)
- Color Layout (*Sikora, 2001*; *Lux & Chatzichristofis, 2008*)
- Edge Histogram (*Sikora, 2001*; *Lux & Chatzichristofis, 2008*)
- Gabor (*Mehrotra, Namuduri & Ranganathan, 1992*; *Lux & Chatzichristofis, 2008*)
- JCD (*Lux & Chatzichristofis, 2008*)
- PHOG (*Dalal & Triggs, 2005*)
- Tamura (*Tamura, Mori & Yamawaki, 1978*)
- LBP (for radius values of 1, 2, 3, 4, 5) (*Ojala, Pietikäinen & Harwood, 1996*; *Liu et al., 2016*; *Liao, Law & Chung, 2009*; *Zhu, Bichot & Chen, 2010*)
- Fine-tuned Light Weight Network (Mobile Net V2) (*Sandler et al., 2018*)

The features that were used for the selection but resulted in the missleading features are as follows:

- Color Histogram (*Lux & Chatzichristofis, 2008*; *Huang et al., 1997*)
- Auto Color Correlation (*Lux & Chatzichristofis, 2008*)
- Speeded up robust features (SURF) (*Lux & Chatzichristofis, 2008*)
- Local Ternary Patterns (LTP) (*Tan & Triggs, 2010*)
- Gray-Level Co-Occurrence Matrix (GLCM) (*Haralick & Shanmugam, 1973*)
- Haralick Features (Statistics of GLCM) (*Haralick & Shanmugam, 1973*)

## Classifications

The final set of features was used for the classification using various neural classifiers. Some of the reasonable classifications resulted from the classification techniques listed below.

**Decision tree classifier**: The decision tree classifier resulted in good accuracy and F1-score in less time. The classifier is used for the various preprocessing steps and for evaluating the results of various features in different combinations. The maximum F1-score of the Decision Tree was 87 for the DowPK dataset and 0.85 for the Kvasir V2 dataset.

**Random forest classifier**: The random forest classifier improved accuracy and F1-score compared to a single decision with a longer detection time. The maximum F1-score was achieved from the dataset of dowPK with an F1-score of 88 while the F1-score for Kvasir V2 dataset was 0.85.

**Fully connected neural network**: The final set of features was used for the detection using a three-layer neural network defined in Eq. (16). The network is trained for 100 epochs to converge to minimum loss.

The approaches improved the detection accuracy, and the detection time was also 41 frames per second (FPS). The final results indicated a commendable F1-score of 0.89 for the Kvasir V1 dataset (*Pogorelov et al., 2017c*). Moreover, the F1-scores for the Kvasir V2 (*Pogorelov et al., 2018*) and Hyper Kvasir (*Borgli et al., 2020*) datasets surpassed 0.90. The approach yielded an impressive F1-score of 0.91 for the dowPK dataset. These promising F1-score values demonstrate the effectiveness of our approach in accurately classifying the GI tract images within these datasets.

## GA-Boost: a solution to tailored class thresholds

Significant variations were found in the probability ranges for different classes. The average probability of images belonging to one class differed from those from other classes. For instance, the images classified as ulcerative colitis exhibited lower probabilities than those classified as polyps. These probability disparities for correctly detected classes highlight the need to utilize different thresholds for each class.

A computational approach based on a genetic algorithm named GA-Boost was employed to address this requirement. GA-Boost algorithm facilitated the generation of optimal thresholds tailored to each class, accommodating the observed variations in probability distributions within the dataset. GA-Boost allowed for the precise adaptation of thresholds for different classes ensuring an effective classification process based on the specific characteristics of each class. The GA-Boost algorithm learned the thresholds for each detected class by applying the genetic algorithm starting with a random threshold ranging from 0.0 to 1.0 with one floating point value. The crossover operator being used in the GA-Boost is the addition with the modulus operator.

The algorithm was executed on a population of 10 probability sets, each comprising 16 randomly generated elements. An additive crossover operator with a mutation rate of 20% was employed. The F1-score evaluation measure was calculated over 20 iterations to select the best chromosomes. As shown in Table 9, the average results of GA-Boost ten runs demonstrated improvements across multiple datasets.

## Statistical analysis

The results were evaluated across ten runs, and a $T$-Test was performed using scipy.stats library. The calculated $p$-values ranged from 0.98 to 0.99 for all evaluation measures on all datasets. As presented in Table 9, the standard deviation of these evaluation measures on the dataset ranged from 0.0001 to 0.001.

Our proposed approach outperforms real-time detection methods regarding detection accuracy, F1-score, and MCC. The results showcase superior accuracy and an impressive detection speed of 41 frames per second (FPS) making it highly efficient for real-time applications.

For a more in-depth comparison of the results across different datasets, please refer to Table 10. This comprehensive analysis highlights the approach's consistent and superior performance in achieving high accuracy while maintaining efficient detection speeds.

**Table 9  Impact on F1-score by applying various additions in the procedure on various datasets.**

| Dataset | Kvasir V1 (*Pogorelov et al., 2017c*) | Kvasir V2 (*Pogorelov et al., 2018*) | Hyper Kvasir (*Borgli et al., 2020*) | DowPK (*Khan, 2023*) |
|---|---|---|---|---|
| No Preprocessing | 0.74 | 0.75 | 0.79 | 0.61 |
| No Augmentation (Reflection Removed) | 0.76 | 0.76 | 0.79 | 0.70 |
| Individual Feature AutoColorCorrelogram | 0.77 | 0.81 | 0.80 | 0.79 |
| Individual Feature ColorLayout | 0.67 | 0.75 | 0.74 | 0.75 |
| Individual Feature EdgeHistogram | 0.59 | 0.64 | 0.70 | 0.60 |
| Individual Feature Gabor | 0.38 | 0.30 | 0.32 | 0.31 |
| Individual Feature JCD | 0.75 | 0.76 | 0.78 | 0.72 |
| Individual Feature PHOG | 0.61 | 0.62 | 0.65 | 0.60 |
| Individual Feature Tamura | 0.49 | 0.48 | 0.52 | 0.42 |
| All Lire Features | 0.71 | 0.82 | 0.82 | 0.79 |
| All Texture Features | 0.75 | 0.88 | 0.88 | 0.89 |
| Deep Features | 0.78 | 0.81 | 0.83 | 0.80 |
| Selected Features (RF-Classifier) | 0.79 | 0.89 | 0.88 | 0.91 |
| 3 Layer Neural Network | 0.89 | 0.90 | 0.90 | 0.91 |
| GA-Boost | 0.90 | 0.91 | 0.91 | 0.93 |

**Table 10  Analysis of LiRE-CNN on various datasets for abnormalities and land mark detection with respect to accuracy and speed.**

| Datasets | Accuracy | F1-score | MCC | Sensitivity | Specificity | AUC-ROC | FPS |
|---|---|---|---|---|---|---|---|
| Kvasir V1 (*Pogorelov et al., 2017c*) | 0.94 | 0.90 | 0.86 | 0.90 | 0.90 | 0.94 | 41 |
| Kvasir V2 (*Pogorelov et al., 2018*) | 0.99 | 0.91 | 0.90 | 0.91 | 0.91 | 0.99 | 41 |
| Hyper Kvasir (*Borgli et al., 2020*) | 0.99 | 0.91 | 0.91 | 0.91 | 0.91 | 0.99 | 41 |
| DowPK | 0.99 | 0.93 | 0.92 | 0.93 | 0.93 | 0.99 | 41 |

Furthermore, we conducted a composite analysis of various algorithmic steps, as detailed in Table 9. The application of Telea reflection removal significantly improved results across most datasets, except for the Hyper Kvasir dataset (*Borgli et al., 2020*). Additionally, the application of feature selection techniques led to varying improvements in the F1-score across, all datasets, demonstrating the effectiveness of these techniques in enhancing the algorithm's performance.

## Results analysis

The application of various additions in the procedure significantly impacted the F1-score across different datasets. The enhancements introduced through these additions resulted in notable improvements in the overall F1-score. The specific effects on the F1-score varied depending on the dataset, indicating the effectiveness of the tailored approach for each

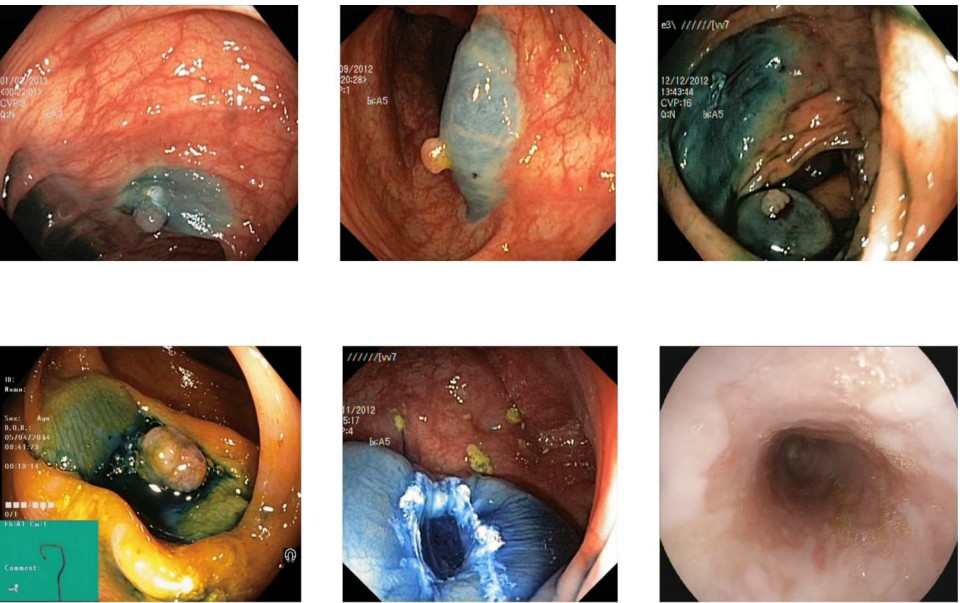

**Figure 7** Some images of various classes from Kvasir V2 (*Pogorelov et al., 2018*) that are correctly detected by the reflection removal while incorrectly decided without preprocessing of reflection removal.

dataset. A detailed analysis of the impact on the F1-score, resulting from applying these additions, is presented in Table 9 dedicated to each dataset.

The reflections on the images misguided the detectors for some images from the actual class to the misleading class. Removing the reflection and using the different thresholds for all the classes led to rectifying some of the images to their correct class. Some of the images resulted in correct detection by applying the reflection removal and different thresholds, shown in Figs. 7 and 8.

There was a problem with the reflections on the image, which led various images to the polyps and dyed classes for some of the non-polyps images. The reflection removal technique improved the polyp detection. The misclassification of other classes to the polyp is reduced with the reflection removal.

There was a class imbalance problem in the datasets that affected the results even after the data augmentation. The accuracy of the minor classes is relatively lower than that of the major classes in all the datasets. The polyp and ulcerative colitis have more images leading to higher accuracy with or without augmentation. There were some classes with very few samples. The class of out-of-patient had only four samples in the Kvasir V2 dataset (*Pogorelov et al., 2018*), which led the model to the over-fitting and the detection of class became 0.40. Only two images of the class are correctly detected out of five test images in the benchmark dataset. The data augmentation improved the accuracy of the minor classes, which affected the complete detection accuracy.

Some of the classes are difficult to differentiate due to the high similarity in the images of those classes. The images of the dyed lifted polyps are very similar to the dyed resection margins because of the same color in both these classes. The images' texture, color, and

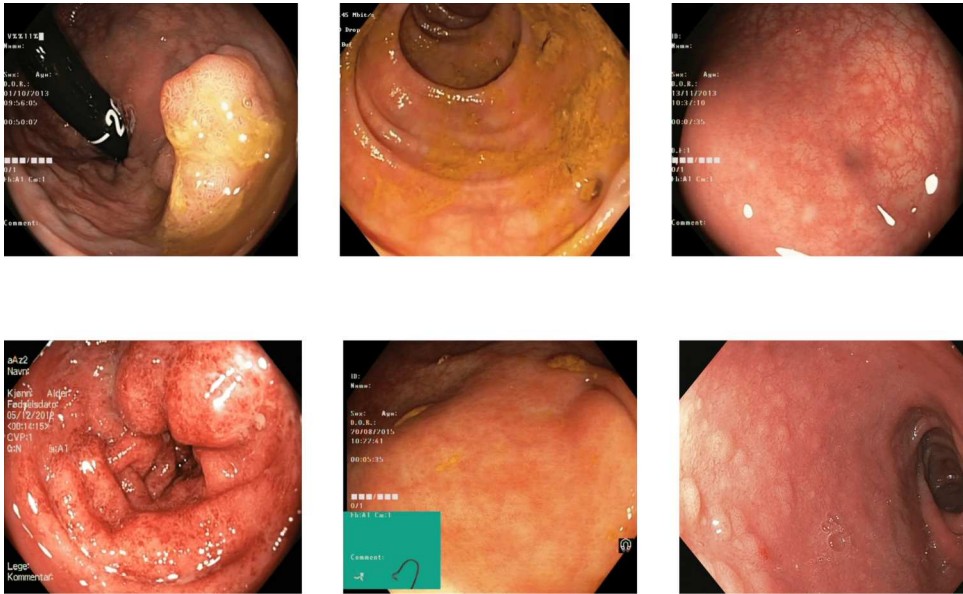

**Figure 8** Some images of various classes from Kvasir V2 (*Pogorelov et al., 2018*) that are correctly detected by the reflection removal and variable threshold.

deep features show similar values for the dyed images of both types, the polyps or resection margins. The detection accuracy between both dyed classes is 0.80 to 0.85 for various benchmark datasets, which can be implemented with a higher focus on these classes. Some misleading images of both types are shown in Fig. 9 with dyed lifted polyps at the top and the dyed resection margins at the bottom.

The other misclassification is found between esophagitis and normal z-line. Esophagitis is an abnormality found in the esophagus, while the z-line is a landmark at the end of the esophagus. Most of the images of esophagitis have the z-line visible, leading to misclassification between these classes. A set of misclassified images from the Kvasir V2 dataset are shown in Fig. 10, with the first row for the Z-index and the second row showing the esophagitis.

The proposed best approach of the LiRE-CNN is compared with the state-of-the-art approaches for the detection of the GI tract abnormalities are shown in Table 11.

## CONCLUSION AND FUTURE WORK

In conclusion, this study comprehensively evaluated diverse methodologies for abnormality and landmark detection. The research delved into two key preprocessing categories: supervised and unsupervised, focusing on reflection elimination. Notably, the most favorable outcomes were attained through unsupervised methods, aligning with dataset requirements and task objectives. Additionally, the challenge of class imbalance was effectively tackled *via* augmentation, utilizing image manipulation encompassing random flips and noise incorporation. These data preprocessing improved performance when applied to several detection techniques. The best technique for fast and correct detection

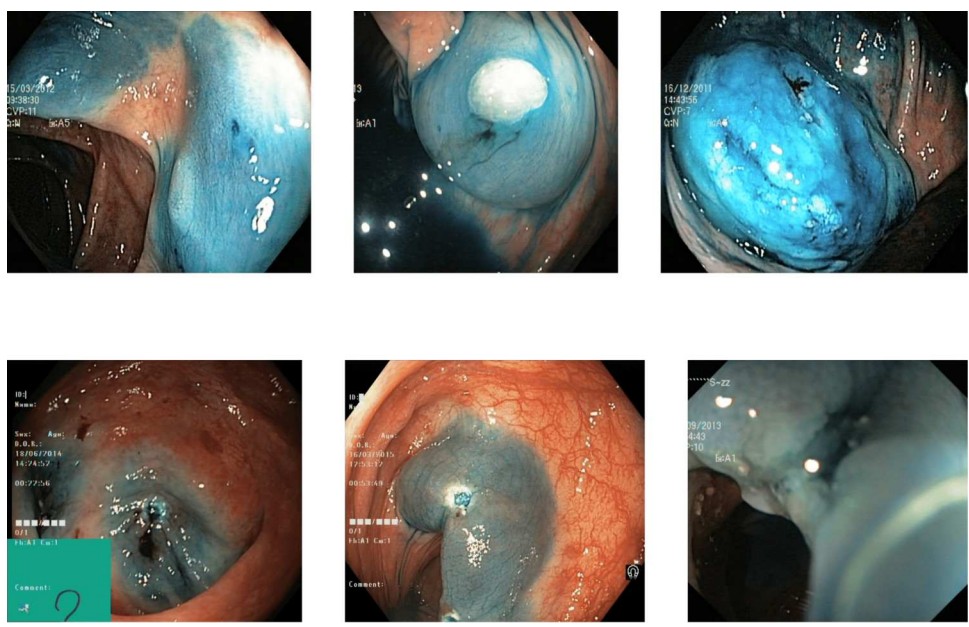

**Figure 9** Incorrectly detected images from the classes of dyed lifted polyps and dyed resection margins in the Kvasir V2 dataset (*Pogorelov et al., 2018*).

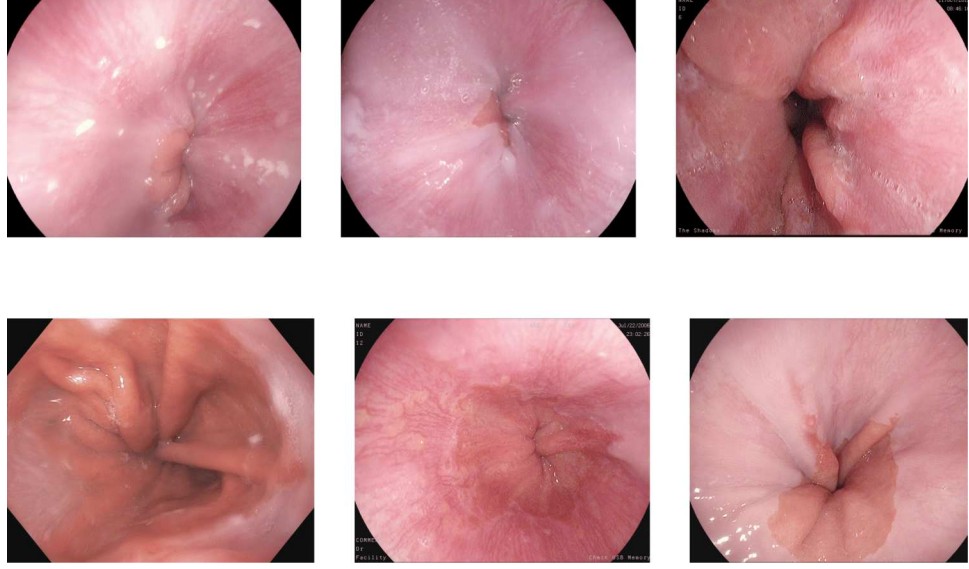

**Figure 10** Incorrectly detected images from the classes of esophagitis and normal z-line in the Kvasir V2 dataset (*Pogorelov et al., 2018*).

**Table 11 Comparison of the best approaches with LiRE-CNN.**

| Sr. No. | Approach | Accuracy | F1-score | FPS |
|---|---|---|---|---|
| 1 | Adaptive Ensembles (*Luo et al., 2019*) | 0.99 | 0.95 | 10 |
| 2 | Combined Neural Networks (*Meng et al., 2019*; *Jha et al., 2021a*) | 0.98 | 0.88 | 98 |
| 3 | Augmentation with Neural Networks (*Hoang et al., 2019*) | 0.99 | 0.95 | 23 |
| 4 | Majority Voting (*Khan & Tahir, 2018*; *Jha et al., 2021a*) | 0.98 | 0.76 | 43328 |
| 13 | Hyper Parameter optimised DenseNet 169 (*García-Aguirre et al., 2022*) | 0.98 | 0.94 | 10 |
| 5 | LiRE-CNN (Proposed) | 0.99 | 0.93 | 41 |

is deep and texture features. The abnormalities and landmarks are classified using a three-layer neural network, resulting in an accuracy of 0.99 and an F1-score of 0.91 with the 41 FPS.

The datasets in the study were only for detecting abnormalities and landmarks. Detecting landmarks and abnormalities can facilitate the treatment of diseases caused by such irregularities by pinpointing and localizing the specific ailment. Some diseases like polyps or cancer needs the segmentation of the affected region for treatment. The future task is to localize the abnormalities and then segment the abnormal region in case of polyps or cancer.

## Funding
This research work was funded by the Higher Education Commission (HEC) Pakistan under NRPU Project 10225/2017. The funders had no role in study design, data collection and analysis, decision to publish, or preparation of the manuscript.

## Grant Disclosures
The following grant information was disclosed by the authors:
Higher Education Commission (HEC) Pakistan: 10225/2017.

## Competing Interests
The authors declare there are no competing interests.

## Author Contributions
- Zeshan Khan conceived and designed the experiments, performed the experiments, analyzed the data, performed the computation work, prepared figures and/or tables, authored or reviewed drafts of the article, and approved the final draft.
- Muhammad Atif Tahir analyzed the data, authored or reviewed drafts of the article, and approved the final draft.

## Data Availability
The code is available at Zenodo: Khan, Zeshan. (2023). RealTime Detection of Endoscopic Abnormalities (V1.2). Zenodo. https://doi.org/10.5281/zenodo.7687070.

The datasets used in the research are available at simula: https://datasets.simula.no/kvasir/.

The dataset is described at KVASIR: A Multi-Class Image Dataset for Computer Aided Gastrointestinal Disease Detection: https://doi.org/10.1145/3193289.

The DowPK dataset is also available at Zenodo:

Zeshan Khan. (2023). DowPK: A dataset generated by the help of endoscopic department of DowPK. https://doi.org/10.5281/zenodo.8343995.

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
