# Peer review of "Real time anatomical landmarks and abnormalities detection in gastrointestinal tract"

_PeerJ Computer Science, doi:10.7717/peerj-cs.1685_

## Round 0.1 · original submission · Major Revisions

Based on the reviewer comments kindly revise the manuscript and resubmit.

Reviewer 1 ·

Basic reporting

Overall, the manuscript titled "Real time anatomical landmarks and abnormality detection in the gastrointestinal tract" presents a robust approach and a well-structured methodology for addressing an important medical problem. There is a lot of potential in the described system. However, there are areas that can be improved to strengthen the manuscript:

Major comments:

- Validity of the reflection removal technique: More justification and explanation of the reflection removal technique in the data preprocessing step are needed. While you mentioned that the technique improves F1-score, it would be helpful to provide more insights on the reason why this step is important and how the reflection affects the detection process.

- Dataset balance: The imbalance of classes in the datasets, particularly in the Kvasir V2 dataset, should be addressed more thoroughly. This imbalance can significantly influence the accuracy, F1-score, and MCC of a model, and it is not clear how this issue was handled in your approach. A robust machine learning model should demonstrate performance across a balanced dataset or otherwise explicitly handle class imbalance.

- Threshold determination via genetic algorithm: The authors mentioned using a genetic algorithm to learn the detection threshold from training data for classes with small inter-class differences and high intra-class differences. However, a detailed explanation or a reference to this methodology is missing. Please include more details or provide an appropriate reference.

- Neural network architecture: The manuscript could benefit from a more comprehensive description or visual representation of the neural network architecture used. Discussing the rationale behind the design choices would be valuable, including why specific activation functions or optimization methods were chosen.

- Performance Evaluation: The presented F1-score, accuracy, and MCC are relatively high, which are great results. However, it would be more convincing to include other metrics such as sensitivity, specificity, or area under the receiver operating characteristic curve (AUC-ROC) which are commonly used in medical image analysis. Also, provide more information about the evaluation strategy used, such as cross-validation, train-test split percentages, and how many runs were averaged to get the presented scores.

- Elimination of Common Calculations: I agree with the comment regarding the calculations of common measures such as accuracy and F1-score. These concepts are widely known and understood, and the detailed calculations can be omitted to simplify the reading process.

- Position of Literature Review: While the Introduction includes some references to previous work, the majority of the literature review seems to be included in the Discussion section. I would suggest moving this to the Introduction section to give readers context for your work and to show how your work differs from and improves upon previous research.

- Language and Writing Style: There are places where the manuscript could benefit from more careful proofreading. For instance, some sentences are a bit long and confusing. Breaking them down into simpler, shorter sentences could improve readability.

Experimental design

no comment

Validity of the findings

no comment

Additional comments

no comment

·

Basic reporting

- The result section is not organized as needed. I recommend authors to redraft the result section in a scientific manner and include more details to make the findings explainable.
- Reflection Removal: I advise authors to provide, in the result section, the number of classes available in each dataset used for reflection removal analysis. It is unclear which classes can be classified with higher accuracy after reflection removal? Also the size training and testing set is missing.
- The results need explanation, why some datasets performed better (F1 score) during Reflection Removal analysis than the others.
- The Augmentation section needs more data. Data augmentation is an essential step for building classification models for clinical diagnosis. Authors mentioned that it significantly increased the detection rate, however Table 8 shows only a minor increase. Are these results (F1 score) replicable during 10-fold CV analysis.
- Line 386, All Features: More details are needed. It's difficult to understand what authors want to convey. It seems to improve the F1 score but the results and associated data is not discussed.
- Line 389, what are those misleading features?
- Line 405, Most distinguishable and conserved features across datasets need to be discussed.
- The article only discusses the strength of methods without involving their limitations. I recommend authors to consider adding limitations too.
- The result section is difficult to understand. I found it confusing what sort of algorithms were used for building models in individual steps and calculating F1 score.
- The objectives of this study mentioned in the introduction (FPM) are not thoroughly discussed and no attempt was made to explain the observations reported in the tables.
- How is this study an improvement over existing studies?

Experimental design

Design and execution of experiment needs major improvement.

Validity of the findings

no comment

Additional comments

For every findings reported in this study, authors need to provide the explanation associated with the observations.

---

## Round 0.2 · accepted · Accept

The author has addressed the reviewer comments properly. Thus I recommend publication of the manuscript.

Reviewer 1 ·

Basic reporting

After reviewing the revised manuscript titled "Real-time anatomical landmarks and abnormality detection in the gastrointestinal tract," I am pleased to see that the authors have addressed all the comments and suggestions provided in the initial review process comprehensively. The revisions add depth to the manuscript and enhance its quality considerably.

Experimental design

No Comments

Validity of the findings

No Comments

Additional comments

No Comments

·

Basic reporting

no comment

Experimental design

no comment

Validity of the findings

no comment

Additional comments

no comment